# The trade-offs of model size in large recommendation models : 100GB to 10MB Criteo-tb DLRM model

**Aditya Desai**
Department of Computer Science
Rice University
Houston, Tx 77005
apd10@rice.edu

**Anshumali Shrivastava**
Department of Computer Science
Rice University/ThirdAI Corp.
Houston, Tx 77005
anshumali@rice.edu

## Abstract

Embedding tables dominate industrial-scale recommendation model sizes, using up to terabytes of memory. A popular and the largest publicly available machine learning MLPerf benchmark on recommendation data is a Deep Learning Recommendation Model (DLRM) trained on a terabyte of click-through data. It contains 100GB of embedding memory (25+Billion parameters). DLRMs, due to their sheer size and the associated volume of data, face difficulty in training, deploying for inference, and memory bottlenecks due to large embedding tables. This paper analyzes and extensively evaluates a generic parameter-sharing setup (PSS) for compressing DLRM models. We show theoretical upper bounds on the learnable memory requirements for achieving approximations to the embedding table. Our bounds indicate exponentially fewer parameters suffice for a good approximation. To this end, we demonstrate a PSS DLRM reaching 10000× compression on criteo-tb without losing quality. Such a compression, however, comes with a caveat. It requires 4.5 × more iterations to achieve the same saturation quality. The paper argues that this tradeoff needs more investigation as it might be significantly favorable. Leveraging the small size of the compressed model, we show a 4.3× improvement in training latency leading to similar overall training times. Thus, in the tradeoff between the system advantage of a small DLRM model vs. slower convergence, we show that scales are tipped towards having a smaller DLRM model, leading to the same quality, faster inference, easier deployment, and similar training times.

## 1 Introduction

Recently, recommendation systems have emerged as one of the largest workloads in machine learning [1]. Recommendation systems form the backbone of a good user experience on online platforms such as e-commerce and web search, where there is a flood of information. Thus, considerable effort goes into building recommendation systems. Deep learning recommendation models give a state-of-the-art performance. However, recommendation models suffer from a critical challenge - sparse features with millions of categorical values[2, 3] These state-of-the-art [2, 4, 5, 6, 7, 8] methods learn a dense representation of the categorical values in a parameter structure called *embedding table*.

Most parameters in recommendation models come from embedding tables. For example, in the popular Criteo-tb MLPerf benchmark model, the embedding tables are around 100GB, whereas other parameters only amount to 10MB. Industrial-scale recommendation models are one of the largest models built, and the size of the embedding table can go as large as hundreds of terabytes. For example, a research article from Facebook discusses training a model of size 50TB over 128 GPUs [3]. The scale of these models leads to some unfavorable effects - slower inference time, slower training time per iteration, and significant engineering challenges in training/deployment. In light of these issues, many works have investigated learning of compressed representation of embedding tables

36th Conference on Neural Information Processing Systems (NeurIPS 2022).

using various principles : (1) compositional embeddings [9] (2) exploiting power-law in the observed frequencies of tokens [10, 11, 12, 13, 14, 15] (3) low-rank decomposition [16], and parameter-sharing methods [17, 18]. Parameter-sharing methods will be the focus of our paper.

We discuss the tradeoffs of parameter-sharing-based compressed recommendation models in terms of quality, inference, and training times. The general idea in machine learning has primarily shifted to larger models and more data. Larger models lead to better capacity, faster convergence, and better generalization. However, the community is realizing that the current route to DL is unsustainable[19]. We must proceed cautiously on this path of building larger models. The large-scale nature of DLRM comes from blown-up embedding parameters - a seemingly inadvertent effect of naive usage of embedding tables. While using large embedding tables gives faster convergence in terms of iterations, the inference is slow and cumbersome. On the other hand, while compressed embedding tables are guaranteed to provide faster inference, it must be seen if they will maintain quality and suffer from slow training times due to slower convergence. In this paper, we make two contributions. First, we theoretically analyze the existence of low-memory PSS for approximating the embedding tables. We find that exponentially fewer parameters are required for good approximations. Secondly, we leverage the system advantage of small models and make hardware-informed implementation choices to combat their slow convergence. We are able to achieve similar training times for compressed recommendation models. In our experiments, we obtain high compression for embedding tables. Additionally, the compressed DLRM models seemingly have no downside in quality and training time while improving inference latency, requiring simple hardware, and reducing costs. We summarize our results below.

We analyze two types of approximations to the embedding table $E : n \times d$. Firstly, we look at a macro-$(\epsilon, \delta)$ PSS approximation which encapsulates the effect of the entire embedding table inside the model. Here, we evaluate the preservation of inner products of the form $\langle Ex, Ey \rangle$, $x, y \in R^d$ under compression. We see that with a memory of $\mathcal{O}(d(d + log(1/\delta))\epsilon^{-2})$, the embedding table can be approximated up to $\epsilon$ relative error with probability $(1 - \delta)$. Secondly, we look at a micro-$(\epsilon, \delta, \rho)$ PSS approximation which looks at how individual embeddings are approximated under compression. Here, we evaluate the preservation of computations of the form $\langle E[i], x \rangle$ $x \in R^d$. We see that with a memory of $\mathcal{O}(d(log(n) + log(1/\delta))\epsilon^{-2})$ we can approximate these computations for important (high norm) embeddings up to an $\epsilon$ relative approximation with probability $(1 - \delta)$. This result motivates us to evaluate the extents of compression that can be achieved for embedding tables. We can obtain 10000× compression (10MB embedding tables) on the criteo-tb dataset to achieve the same target quality - an order of magnitude higher compression than previously reported. This small model improves inference time, eliminates engineering challenges, and makes the models easy to deploy on low-resource devices. However, as expected, this 10 MB sized embedding model requires 4.5 epochs to train.

Consistent with a general observation in machine learning that the number of iterations required to converge decreases with an increase in parameters, the compressed recommendation models suffer from slow convergence. We observe that the higher the compression, the slower the convergence. For instance, a 1000× compressed DLRM model requires 1.9 epochs to converge, while a 10000× compressed model requires 4.5 epochs to converge. This is worrisome as training times are one of the significant aspects of the model. However, a highly compressed model completely changes the storage of model parameters (e.g., multiple nodes to a single node, RAM to GPU, etc.). Hence, we observe a steep decrease in the latency of embedding lookup/gradient update operations. An illustration of this tradeoff is presented in figure 1. We investigate how this improvement in latency compensates for increased training iterations. Unfortunately, existing parameter-sharing methods do not demonstrate much training time per iteration improvement due to various reasons such as high algorithmic complexity, poor cache efficiency, and sub-optimal implementation strategies not cognizant of the underlying hardware. We evaluate various implementation choices and demonstrate that the advantage of faster training time per iteration almost compensates for the slower convergence in compressed models. For instance, for a 1000× compressed model, we can lower the training latency by 1.7×, making the overall training time only 1.1× original model. Similarly, we can improve the training latency 4.3× for a 10000× compressed model making the overall training time 1.04×

## 2 Parameter-shared embedding tables

In this section, we analyze the parameter-shared embedding tables. We start with a few definitions.

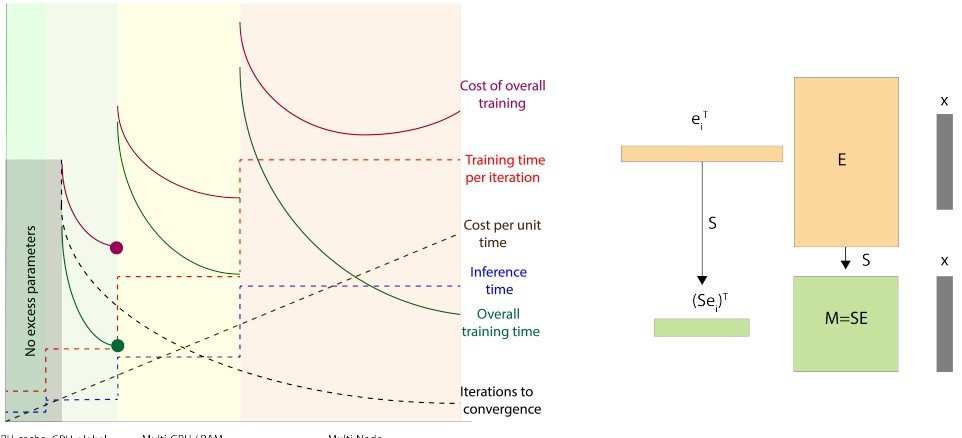

Figure 1: **(left)**This not-to-scale illustration shows tradeoffs of having larger parameters ( and hence the memory footprint of the model ). Each line has its scale. As we move towards the right, memory increases, and we need to have memory farther away from computational hardware. This increases the latency and, thus, inference time and training time per iteration. The energy cost also increases due to larger active hardware. We observe that the convergence becomes faster as the number of parameters increases. The solid lines show the overall training time and costs, showing that more parameters are not always better. **(right)** An Illustrating the JLT-based setup of PSS, the computation involving E is performed in the sketched space using much lesser memory of learnable parameters M. The sketching is achieved using sketching matrix S

**Definition 1** (**Parameter shared setup**). *Under parameter shared setup (PSS) for embedding table of size $n \times d$, we have a set of weights $M$ (usually $|M| \ll nd$) and recovery function $\mathcal{M}$ : $R^{|M|} \times \{0, .., n-1\} \to R^d$. The embedding of a token $i \in \{0, .., n-1\}$ is recovered as $\mathcal{M}(M, i)$.*

M can be represented with any memory layout, for instance, a 2D or 1D array. We denote the size of $M$ as $|M|$ and memory to store $\mathcal{M}$ as $|\mathcal{M}|$. $\mathcal{M}$ shares the weights in $M$ to generate the entire embedding table. Note the inherent trade-off between $|M|$ and $|\mathcal{M}|$. We can make $M$ large in order to get a very simple $\mathcal{M}$ (for example, $M=E$ and $\mathcal{M}(M, i)=M[i]$) or we can keep $M$ very small at the cost of complicated $\mathcal{M}$ (for example $M = [0,1]$ and $\mathcal{M}$ combines these bits to get bit representation of $E$). An effective PSS for embedding table $E \in R^{n \times d}$ would have $(|M| + |\mathcal{M}|) \ll nd$. Also, for efficient training and inference, we want $\mathcal{M}(M, i)$ computation to be cheap. Training a PSS involves directly training M as a part of end-to-end training. However, the initial part of the discussion will focus on approximating a ground truth embedding table $E$.

One of the key questions we want to answer is how small a PSS we can create for a ground truth embedding table $E$. That is, how small can $(|M| + |\mathcal{M}|)$ be. For an arbitrary $E$, it is impossible to accurately encode $E$ in sublinear memory. So, next, we define a $(\epsilon, \delta)$ approximate PSS. While creating an alternative representation of embedding table $E$, let us see what we would like to achieve. First, we would like to preserve all the pairwise inner products between two embeddings of tokens from the embedding table. Additionally, we also want to minimally affect the subsequent computations performed on the embeddings retrieved from the embedding table. In recommendation models, the embeddings retrieved are passed through neural network layers, which will perform inner products on embeddings. Based on this, we analyze two types of $(\epsilon, \delta)$-PSS.

**Definition 2** (**Macro $(\epsilon, \delta)$-PSS**). *In cases where $Mx, x \in R^d$ is well defined and can be interpreted as compressed version of $Ex$, a PSS $(M, \mathcal{M})$ is a Macro $(\epsilon, \delta)$-PSS for an embedding table E if the following holds with probability $(1 - \delta)$,*

$$\forall x, y \in R^d \quad |\langle Mx, My \rangle - \langle Ex, Ey \rangle| \le \epsilon \|x\|_2 \|y\|_2$$

The Macro-PSS captures the effect of the entire embedding table along with the model. Consider the set of all vectors $Ex, x \in R^d$. Under our PSS compression, we want the geometry of this space of vectors to be approximately preserved. Hence, in this definition, we preserve the inner products between all the vectors of the form $Ex, x \in R^d$.

**Definition 3** (**Micro** $(\epsilon, \delta, \rho)$**-PSS**). *A PSS* $(M, \mathcal{M})$ *is a Micro* $(\epsilon, \delta, \rho)$*-PSS for an embedding table E if the following holds with probability* $(1 - \delta)$,

$$\forall x \in R^d \; \forall i \in \{0, .., n-1\}, \; \text{s.t} \; \rho(i) > \rho \quad |\langle \mathcal{M}(M, i), x \rangle - \langle E[i], x \rangle| \le \epsilon \|E[i]\|_2 \|x\|_2$$

*where* $\rho(i)$, *the importance factor for embedding* $i$, *is defined by*,

$$\rho(i) = \frac{\|E[i]\|_2}{\sigma(E)}$$

*where* $\sigma(E)$ *is the maximum singular value of the embedding matrix.*

The Micro-PSS captures the effect of compression on each embedding on the model individually. Note that it is impossible to guarantee good recovery for all the embeddings from the embedding table. Thus, we define Micro-PSS with an additional parameter $\rho$, which measures the importance threshold of embeddings. As we will see, we can guarantee good recovery in logarithmic memory for embeddings with importance (as defined in the above definition) higher than $\rho$.

Interestingly, the Micro $(\epsilon, \delta, \rho)$-PSS for embedding table E also preserves the pairwise inner products between the embeddings of different tokens, as is mentioned in the following result.

**Theorem 4.** *Let* $(M, \mathcal{M})$ *be a Micro* $(\epsilon, \delta, \rho)$*-PSS for embedding table E, we have ,*

$$\forall i, j \in \{0, ..., n-1\}, \; \text{s.t} \; \rho(i), \rho(j) > \rho$$
$$|\,|\langle \mathcal{M}(M, i), \mathcal{M}(M, j) \rangle - \langle E[i], E[j] \rangle| \le (\epsilon^2 + 2\epsilon) \|E[i]\|_2 \|E[j]\|_2$$

*Proof.* Presented in the appendix $\qquad\qquad\qquad\qquad\qquad\qquad\qquad\qquad\qquad\qquad$ □

Micro-PSS caters to the practical embedding tables in which a few embeddings are much more important than a long tail of non-important embeddings. In our formulation, we characterize this importance by the norm of the embeddings. It has found different manifestations in literature. For instance, mixed-dimensional embeddings [10, 11, 12, 13, 14, 15] have longer embeddings and tensor-train recommendation have higher rank for important tokens. In existing methods, we have to pre-identify the important tokens, which is usually done by looking at the frequencies of their occurrence in data. On the contrary, in PSS, this happens naturally, and we do not have to identify these tokens beforehand.

### 2.1 $(\epsilon, \delta)$-PSS via Johnson–Lindenstrauss Transforms (JLT)

This section is dedicated to analyzing the existence of $(\epsilon, \delta)$-PSS for embedding tables leveraging JLT matrix-based sketching. We will use the following definition of JLT from [20]

**Definition 5.** *A randomly generated matrix* $S \in R^{k \times n}$ *is a* $JLT(\epsilon, \delta, f)$ *if with probability at least* $(1 - \delta)$, *for any* $f$*-element subset* $V \subset R^n$ *the following holds,*

$$\forall v_1, v_2 \in V \; |\langle S v_1, S v_2 \rangle - \langle v_1, v_2 \rangle| \le \epsilon \|v_1\|_2 \|v_2\|_2$$

JLT matrices directly give us a bound on the number of parameters required for a Macro $(\epsilon, \delta)$-PSS. It is stated in the following theorem,

**Theorem 6.** *Let the embedding table be* $E \in R^{n \times d}$. *Consider a matrix S which is* $JLT(\epsilon, \delta, 9^d)$). *Then* $M$ *and* $\mathcal{M}$ *defined by*

$$M = SE \quad \mathcal{M}(M, i) = (Se_i)^\top (M)$$

*is a Macro*$(\epsilon, \delta)$*-PSS.*

*Proof.* The property of S that we are interested in is called l2-subspace embeddings[20]. We leverage the result from [20] that $JLT(\epsilon, \delta, 9^d)$ is a l2-subspace embedding matrix. More details are in the appendix $\qquad\qquad\qquad\qquad\qquad\qquad\qquad\qquad\qquad\qquad\qquad\qquad\qquad\qquad\qquad\qquad$ □

Using independent standard normal variables to instantiate S[20], we can construct a $JLT(\epsilon, \delta, 9^d)$ matrix S of size $k \times n$ where $k = \Omega((d + log(1/\delta))\epsilon^{-2})$. Thus the model size $|M| = |SE|$ is $kd = \Omega(d(d + log(1/\delta))\epsilon^{-2})$. Thus, with JLT matrix sketching, we can obtain a Macro-$(\epsilon, \delta)$-PSS in space $|M| = \mathcal{O}(d^2 \epsilon^{-2})$. This might be an explanation as to why we can achieve high compression

in embedding tables. We will discuss the storage cost and execution of $\mathcal{M}$ shortly. However, we want to note here that we can relax the restrictions on the distribution of S so that the storage cost of $|\mathcal{M}| = \mathcal{O}(1)$ and the cost of applying $\mathcal{M}$ is $\mathcal{O}(kd)$

The Macro-PSS does not discuss the individual embedding vector quality; hence, we analyze the Micro-PSS for embedding tables in the following theorem.

**Theorem 7.** *Let the embedding table be $E \in R^{n \times d}$. Consider a matrix S which is $JLT(\rho\epsilon, \delta, 9^d + n)$. Then $M$ and $\mathcal{M}$ defined by*

$$M = SE \quad \mathcal{M}(M, i) = (Se_i)^\top (M)$$

*is a Micro $(\epsilon, \delta, \rho)PSS$ where $e_i \in R^n$ is a one-hot encoding of integer $i$ (i.e $e_i[i] = 1$ and rest all elements of $e_i$ are 0).*

*Proof.* The complete proof is presented in the appendix. The intuition is that we want to maintain the computations of the form $e_i^\top Ex$, which can be seen as an inner product between $e_i \in R^n$ and $Ex \in R^n$. So the inner products between vectors of sets $A = \{e_0, ..., e_{n-1}\}$ and column space of $E$ need to be preserved. Once we observe this, we apply JLT to the set of n discrete points and the $1/2 - net$ of column space of $E$, which contains less than $9^d$ points, and get our result. $\square$

Using S constructed by standard normal variables [20], we can construct a $JLT(\epsilon\rho, \delta, 9^d + n)$ matrix S of size $k \times n$ where $k = \Omega((log(9^d + n) + log(1/\delta))\rho^{-2}\epsilon^{-2})$. Thus the model size $|M| = |SE|$ is $kd = \Omega(d((log(9^d + n) + log(1/\delta))\rho^{-2}\epsilon^{-2}))$. Thus, we can obtain a Micro-$(\epsilon, \delta, \rho)$-PSS in space $\mathcal{O}(\max(d, log(n))d\epsilon^{-2}\rho^{-2}))$. Next, we will discuss the storage and execution cost of $\mathcal{M}$.

**Storage of JLT matrices and relaxations.** In using PSS in end-to-end training, we would learn the memory $M$ while keeping the mapping $\mathcal{M}$ constant. Thus, unlike in sketching for linear algebra [20], we care about the costs of (1) storage of learnable parameters $|M|$, (2) Cost of storing mapping $|\mathcal{M}|$, and (3) cost of computing $\mathcal{M}(M, i)$. It is clear that using standard normal JLT is not feasible due to storage of matrix S ($= kn$) will be more expensive than storing $E$ itself! There are a lot of sparse JLT$(\epsilon, \delta, f)$ matrices S which will reduce the storage cost of $S$ [21, 22, 23]. [23] showed that the matrix needs to have a minimum column sparsity of $\Omega(\epsilon^{-1}log(f/\delta)log(1/\epsilon))$ and hence the cost of storing S while using JLT matrices is lower bounded by $\Omega(n\epsilon^{-1}log(f/\delta)log(1/\epsilon))$ which can still be considerably high. However, the issue with storage costs of S can be alleviated in practice by relaxing the complete independence condition on entries of S. For example, the JLT matrix (which requires the same bound on k as given in theorem 7) suggested by [21], which selects each entry to be 0 with probability 2/3, ±1 with probability 1/6 independently can be generated on-the-fly using universal hashing. [24] analyze why simpler hash functions can work well with data having enough entropy. The benefit of using universal hashing is that mapping $\mathcal{M}$ can now be stored in $O(1)$ memory. Thus using the [21] sparse sketching with relaxed independence, we can store $JLT(\epsilon, \delta, f)$ matrix in $O(1)$ memory. Thus, total memory for Macro-PSS would be $|M| + |\mathcal{M}| = \mathcal{O}(d^2\epsilon^{-2})$, whereas, total memory for Micro-PSS would be $|M| + |\mathcal{M}| = \mathcal{O}(d(\max(d, log(n)))\epsilon^{-2}\rho^{-2})$. In both cases, the cost of applying mapping $\mathcal{M}$ to recover an embedding is $\mathcal{O}(kd)$.

### 2.2 Training end-to-end $(\epsilon, \delta)$ PSS for embedding table E

The above discussion directly gives us an algorithm to compute a $(\epsilon, \delta)$ PSS from a trained embedding table $E$. We just need to compute $M = SE$ and while retrieving an embedding compute $\mathcal{M}(M, i) = (Se_i)^\top M$. We can also directly train an $(\epsilon, \delta)$ PSS in an end-to-end manner. The idea is to train the compressed $M$ directly. Thus, we have a matrix of learnable weights $M \in R^{k \times d}$. Let us now look at how the forward and backward pass of embedding retrieval mapping $\mathcal{M}$ looks like

$$\text{forward(i)} = (Se_i)^\top M$$

The forward function takes an integer $i$ and returns a $R^{1 \times d}$ array which is embedding of $i$.

$$\text{backward(i}, \Delta) = (Se_i)\Delta$$

The backward pass takes in all the arguments of the forward pass along with the $\Delta \in R^{1 \times d}$, which are gradients of loss with respect to the output of the forward pass. We can back-propagate further to $W$ using the above formulation. The result of the backward pass is a $k \times d$ matrix.

If we use a sparse $S$ such as with sparse JLT, we can achieve sparse gradients of W. That is, only a few W's gradient entries are non-zero. As we will see in section 3, whether to propagate sparse or dense gradients is a vital implementation choice. We were able to achieve better training per iteration by exploiting this choice.

Table 1: State of the art in embedding compression on criteo datasets. * are not PSS )

| Method | Dataset | Compression | Quality |
|--------|---------|-------------|---------|
| HashingTrick | Crieto Kaggle | 4× | worse |
| QR Trick | Criteo Kaggle | 4× | similar/slightly worse |
| MD Embedding* | Criteo Kaggle | 16× | better/similar |
| TT-Rec* | Criteo kaggle/Criteo TB | 112× / 117 × | better/simiar |
| ROBE | Criteo Kaggle/Criteo TB | 1000× | better/similar |

## 2.3 Practical PSS and existing SOTA methods

Existing parameter-sharing based embedding compression methods can also be seen as PSS with varied distributions over S. We state a few of them below.

- **Hashing Trick** In this method, entire embeddings for a token $i$ is drawn from a randomly hashed location. In terms of PSS, we can define the mapping function $\mathcal{M}_h$ over a 2D matrix $\mathcal{M}_h \in R^{k \times d}$ where $k < n$.

$$\mathcal{M}_h(M_h, i)[:] = M_h[h(i), :]$$

  where h is a hash function.

- **QR decomposition**[9]. In this method, embedding for a token $i$ is drawn in chunks from separate memory vectors. In terms of PSS, we can define the mapping function $\mathcal{M}_q$ over, say l, pieces of 2D memory $M_1, M_2, ..M_l \in R^{k \times d/l}$ as follows

$$\mathcal{M}_q(\{M_1, M_2, ..M_k\}, i)[j * (d/l) : (j + 1) * (d/l)] = M_j[h_j(i), :] \tag{1}$$

  In words, the $j^{th}$ chunk of $i^{th}$ embedding is recovered using the chunk at the location $h_j(i)$ in memory $M_r$

- **HashedNet [17] and ROBE-Z [18]** : In HashedNets, authors proposed mapping model weights randomly into a parameter array. ROBE-Z extended this idea by hashing chunks of embedding vector instead of individual elements. In terms of PSS, we can define the mapping function $\mathcal{M}_r$ over 1D memory $M_r$ as

$$\mathcal{M}_r(M_r, i)[j * Z : (j + 1) * Z] = M_r[h(i, j) : h(i, j) + Z]$$

  That is, the $j^{th}$ chunk of $i^{th}$ embedding is recovered using the chunk at the location $h(i, j)$ from $M_r$. Here, $h : \mathbf{N}^2 \rightarrow \{0, .., |M_r|\}$ is a hash function. If we set $Z = 1$, then we get the mapping function for HashedNet.

We can summarize the state-of-the-art embedding compression in the table 1. Our Micro-PSS theory suggests that we should be able approximate the embedding table in memory of the order of $\mathcal{O}(\log(n))$. Thus, it is important to question if 1000× compression is the best compression we can achieve on a dataset like CriteoTB where we have large (~400GB sized) embedding tables. As we shall see, one of the significant roadblocks in aiming to achieve higher compression is training time. The higher the compression, the more iterations are needed to train the model to a certain quality. In the next section, we discuss how to overcome this bottle-neck preventing training of highly compressed PSS.

## 3 Road to 10000× compression with PSS

Table 3 shows the quality of embeddings for the criteo-kaggle dataset over varying compression rates across different popular recommendation models. We can see that the state-of-the-art compression of the criteo-kaggle dataset is 1000×. It is a consequence of PSS theory that larger embedding tables (prevalent in industrial scale recommendation models), such as those for criteo-tb, should obtain more compression. We aim to study higher compression for large embedding tables and improve over existing SOTA (1000x).

A significant roadblock in building highly compressed embedding tables is the slow convergence. Table 3 shows that the number of iterations required to converge increases with higher compression across different deep-learning recommendation models. Specifically, for 1000× compression, it takes up to 4× iterations, and for models with 10000× compression, most models do not converge in 15 epochs. If a model does not converge, it is hard to judge if the convergence is too slow or if its capacity is consumed. In order to be able to train for larger iterations, we should be able to train the highly compressed models faster. Indeed, we expect some system benefits with highly compressed models,

Table 2: These experiments are run on a single GPU for a simple embedding lookup and loss is taken to be sum of all retrieved elements with a batch size of 10240 and embedding dimension=128. ■ higher latency (worse) ■ lower latency (better)

| | | 4M | 8M | 16M | 32M | 64M | 128M | 256M | | | 4M | 8M | 16M | 32M | 64M | 128M | 256M |
|---|---|---|---|---|---|---|---|---|---|---|---|---|---|---|---|---|---|
| | | (a) forward pass | | | | | | | | | (d) forward pass | | | | | | |
| | $n \longrightarrow$ | 4M | 8M | 16M | 32M | 64M | 128M | 256M | | | 4M | 8M | 16M | 32M | 64M | 128M | 256M |
| compression | 10× | 0.40 | 0.36 | 0.37 | 0.37 | 0.37 | 0.37 | | chunk size | 1 | 0.68 | 0.75 | 0.77 | 0.80 | 0.92 | 0.88 | 0.88 |
| | $10^2\times$ | 0.31 | 0.32 | 0.31 | 0.25 | 0.37 | 0.37 | 0.36 | | 4 | 0.55 | 0.54 | 0.54 | 0.54 | 0.65 | 0.65 | 0.65 |
| | $10^3\times$ | 0.27 | 0.29 | 0.30 | 0.31 | 0.30 | 0.32 | 0.31 | | 16 | 0.31 | 0.31 | 0.31 | 0.24 | 0.37 | 0.37 | 0.36 |
| | $10^4\times$ | 0.27 | 0.27 | 0.27 | 0.26 | 0.28 | 0.29 | 0.30 | | 32 | 0.31 | 0.31 | 0.31 | 0.23 | 0.37 | 0.37 | 0.36 |
| | | (b) backward pass (sparse = false) | | | | | | | | | (e) backward pass (sparse = false) | | | | | | |
| | 10× | 2.55 | 4.81 | 9.15 | 17.81 | 35.10 | 69.73 | | | 1 | 1.29 | 1.56 | 2.02 | 2.96 | 4.69 | 8.15 | 15.17 |
| | $10^2\times$ | 0.59 | 0.83 | 1.27 | 2.13 | 3.85 | 7.42 | 14.34 | | 4 | 0.80 | 1.07 | 1.54 | 2.40 | 4.13 | 7.70 | 14.61 |
| | $10^3\times$ | 0.35 | 0.37 | 0.41 | 0.54 | 0.73 | 1.08 | 1.78 | | 16 | 0.58 | 0.82 | 1.27 | 2.12 | 3.85 | 7.43 | 14.34 |
| | $10^4\times$ | 0.35 | 0.35 | 0.34 | 0.35 | 0.36 | 0.40 | 0.47 | | 32 | 0.57 | 0.80 | 1.27 | 2.12 | 3.85 | 7.42 | 14.34 |
| | | (c) backward pass (sparse = true) | | | | | | | | | (f) backward pass (sparse = true) | | | | | | |
| | 10× | 2.67 | 2.53 | 2.52 | 2.51 | 2.51 | 2.52 | 2.54 | | 1 | 3.17 | 3.41 | 3.56 | 3.68 | 3.94 | 3.98 | 4.00 |
| | $10^2\times$ | 2.48 | 2.58 | 2.68 | 2.64 | 2.66 | 2.65 | 2.52 | | 4 | 2.61 | 2.64 | 2.74 | 2.73 | 2.91 | 2.77 | 2.74 |
| | $10^3\times$ | 2.56 | 2.48 | 2.58 | 2.56 | 2.63 | 2.65 | 2.53 | | 16 | 2.50 | 2.66 | 2.54 | 2.53 | 2.52 | 2.51 | 2.53 |
| | $10^4\times$ | 2.52 | 2.53 | 2.45 | 2.55 | 2.60 | 2.54 | 2.62 | | 32 | 2.48 | 2.61 | 2.62 | 2.50 | 2.64 | 2.60 | 2.50 |

as shown in figure 1. We find that existing methods cannot train highly compressed embedding tables because of the implementation choices. We propose the following implementation choices.

**Hashing chunks is better than hashing elements:** As suggested in [18], we find that hashing chunks instead of individual elements is indeed helpful in reducing the latency of PSS systems. As shown in table 2(d,e), both forward and backward passes are affected a lot by chunk sizes. It is also interesting to note that the effect of chunk sizes diminishes after size 16. Hence, we can choose a chunk size of 32 for hashing.

**Dense gradients are suitable for highly compressed PSS** Generally, in embedding tables, implementations use sparse gradients. Dense gradients are particularly wasteful in embedding tables, where in each iteration, only a few weights are involved in the computation. However, the same does not apply to highly compressed PSS. In the case of PSS, the algorithm for sparse gradient computation is more involved than that for dense gradients. The algorithms for sparse and dense gradient computation are specified in algorithm 1. Also, the scatter operation is much faster when the memory size of embeddings is smaller. Hence, dense gradients work well with higher compressions. From table 2(b,c), we can see that at higher compression rates, the backward pass is up to 8× faster with dense gradients as compared to sparse gradients.

---

**Algorithm 1** Backward Pass for PSS

---

**Require:** $\mathcal{G}_O \in R^{B \times d}, \mathcal{I} \in R^{B \times d}$, sparse $\in \{\text{True}, \text{False}\}$, M: compressed memory.
        ▷ $\mathcal{G}_O$ is the gradient w.r.t output, and $\mathcal{I}$ is the mapping matrix showing the locations from which the output was accessed
**Ensure:** $\mathcal{G}_i$ is gradient w.r.t input.
  **if** sparse **then**
      unique, invIdx $\leftarrow$ findUnique($\mathcal{I}$.flatten(), returnInverse = True)
      $g \leftarrow$ Tensor$((\text{unique.shape}, ))$
      $g$.scatter_add(invIdx, $\mathcal{G}_o$.flatten()
      $\mathcal{G}_i = $ sparseTensor(location = unique, values = g, size = M.size)
  **else if** dense **then**
      $\mathcal{G}_i \leftarrow$ Tensor(M.size)
      $\mathcal{G}_i$.scatter_add(0, $\mathcal{I}$.flatten(), $\mathcal{G}_o$.flatten())
  **end if**

---

**Forward/Backward kernel optimizations** Latency for PSS, and CUDA kernels in general, is very sensitive to the usage of shared memory, occupancy, and communication between CPU and GPUs. We also optimize our PSS code to minimize the data movement costs, implement custom kernels to fuse operations to improve shared memory usage, and optimize CUDA grid sizes to obtain the best performance.

With the implementation choices mentioned above, we are ready to train a 10000× compressed PSS for criteo-tb. We implement our PSS using ROBE-style hashing.

Table 3: **Criteo-kaggle**: Quality and convergence on of 5 popular models vs. compression. The standard deviation of all AUC results is within 0.0009. **(left)** Quality of models for criteo-kaggle dataset is largely maintained till 1000x compression after which either the convergence is too slow / capacity of model drops (max 15 epochs) **(right)** Epochs needed to converge to target AUC set to minimum of value achieved by original model. (max 15 epochs)

| | Quality of models (15 epochs max) | | | | | Epochs to reach target AUC | | | | | |
|---|---|---|---|---|---|---|---|---|---|---|---|
| | ORIG | 10× | $10^2\times$ | $10^3\times$ | $10^4\times$ | Target AUC | ORIG | 10× | $10^2\times$ | $10^3\times$ | $10^4\times$ |
| DLRM | 0.8031 | 0.8032 | 0.8029 | 0.8048 | 0.8001 | 0.8029 | 1.00 | 1.52 | 1.78 | 2.78 | - |
| DCN | 0.7973 | 0.7982 | 0.7978 | 0.7991 | 0.7967 | 0.7973 | 1.00 | 1.00 | 1.87 | 1.93 | - |
| AUTOINT | 0.7968 | 0.7972 | 0.7968 | 0.7987 | 0.7957 | 0.7968 | 1.00 | 1.00 | 2.00 | 2.50 | 14.93 |
| DEEPFM | 0.7957 | 0.7961 | 0.7953 | 0.7951 | 0.7943 | 0.7951 | 1.00 | 1.00 | 1.00 | 3.93 | - |
| XDEEPFM | 0.8007 | 0.8016 | 0.7998 | 0.7989 | 0.795 | 0.7989 | 1.62 | 1.50 | 1.74 | 3.93 | - |
| FIBINET | 0.8016 | 0.8021 | 0.8011 | 0.8011 | 0.7963 | 0.8011 | 1.00 | 0.93 | 1.00 | 2.99 | - |

# 4 Experimental results on Criteo datasets

We perform experiments on criteo-kaggle and criteo-tb datasets in order to confirm the following hypothesis,

1. Theory in section 2 dictates that important embeddings in table ($E \in R^{n \times d}$) can be represented in memory logarithmic in $n$. High compression should be possible in these datasets. Specifically, higher compression should be possible in larger embedding tables.
2. The system advantage of smaller PSS should compensate for the convergence advantage of the original model.

**Datasets:** Criteo-kaggle and criteo-tb datasets have 13 integer and 26 categorical features. criteo-kaggle data was collected over seven days, whereas the criteo-tb dataset was collected over 23 days. criteo-tb is one of the largest recommendation datasets in the public domains with around 800 million token values making the embedding tables of size around 400GB. (with d = 128). criteo-kaggle is smaller and has a 2 GB-sized embedding table. Note that industrial-scale models are much larger than the DLRM model we talk about here. For example, Facebook recently published a model sized 50TB [3]. One can extrapolate the benefits of PSS to industrial-scale models based on this case study.

**Models:** Facebook MLPerf DLRM[2] model, available under Apache-2.0 license, for the criteo-tb dataset, achieves the target AUC (0.8025) with the embedding memory of around 100GB. This model uses a maximum cap of 40M indices per embedding table, leading to a total of 204M embeddings. This model cannot be trained on a single GPU (like V100) and is trained using multiple GPUs (4 or 8). For the criteo-tb dataset, we use the DLRM MLPerf model. For criteo-kaggle dataset, we use an array of state-of-the-art models DLRM[2], DCN[4], AUTOINT[5], DEEPFM [6], XDEEPFM [8] and FIBINET. We use our PSS implementation described in 3 as a compressed model.

**Quality of model vs. Excess parameters:** Tables 3(b) and 4(b) show the results for the two datasets across different values of compression. In table 3(b), we can see that across different models, the quality of the model is maintained until 1000× compression. As criteo-tb embedding tables are much larger than the criteo-kaggle dataset, according to the section 2, we should see larger values of compression. Indeed, we obtain a 10000× compression without loss of quality of the model for criteo-tb. This level of compression is unprecedented in embedding compression literature. These experiments validate our first hypothesis and provide a new state-of-the-art embedding compression. In the rest of the section, we evaluate how the system advantage of PSS compares against the original embeddings in various aspects.

**Inference time** In figure 2a, we compare the inference time of test data (89M samples) in the criteo-tb dataset with a batch size of 16384 with PSS on a single Quadro RTX-8000 GPU and the original model on 8 Quadro RTX-8000 GPUs. The total time for inference for the original model is around 203 seconds, whereas PSS is around 97 seconds. In the original model trained on 8GPUs, embedding lookup is model-parallel, whereas the rest of the computation ( bot-mlp, top-mlp, and interactions) is data-parallel. There is a steep improvement of over 3× in embedding time lookup as we move from a distributed GPU setup for embedding tables to a PSS on a single GPU. The extra computation time in PSS is much smaller than all-to-all communication costs in the original embedding table lookup. We include the time required for data distribution (initial) in bot-mlp and data-gather (final) in top-mlp timings. This communication cost is high, and we can see that we are better off performing the entire computation on a single GPU.

Table 4: **Criteo-tb** : Quality and convergence on criteo-tb with DLRM vs. compression. The standard deviation of AUC results is within 0.0009. (1) $10^4\times$ compression also reaches the target AUC. For $10^5\times$ convergence is too slow/mode capacity is reached. (max 15 epochs) } (2) Epochs required to reach 0.8025 AUC for DLRM model on criteo-tb dataset. Relative time computes the ratio of A*B to original model

|  | ORIG | $10^2\times$ | $10^3\times$ | $10^4\times$ | $10^5\times$ |
|---|---|---|---|---|---|
| Target reached | Yes | Yes | Yes | Yes | No |
| Epochs to reach target (A) | 1.00 | 1.94 | 1.9 | 4.55 | - |
| Time / 1000 iteration (B) | 50.6 | 31.04 | 29.6 | 11.6 | - |
| Relative Time (C = A B) | 1 | 1.19 | 1.11 | 1.04 | - |

**Model training time per iteration vs excess parameters** Figure 2b records the time-taken for 1000 iterations of training. While using embedding tables, we can choose to back-propagate sparse or dense embedding gradients. The general idea is to use sparse gradients when very few gradients are non-zeros. In the original model, we can only use sparse gradients as using dense gradients is prohibitive w.r.t to computation and communication. In PSS, however, we compare both modes for gradient back-propagation. We see that dense gradients perform exceptionally well at higher rates of compression. The performance of sparse gradients is constant across different compression rates as the workload is similar. Generally, it seems good to use dense gradients for PSS when the effective memory size (i.e., the final memory of compressed embedding tables ) is small. It is noteworthy that the training time per iteration reduces significantly with higher compression. For example, with 1000x compression, the training time is $1.7\times$ lesser than the time taken by the original model, whereas with 10000x compression, the time per iteration is $4.37\times$ lesser.

**Model convergence and overall time vs. excess parameters** We observe a consistent trend in convergence: as the number of excess parameters in the models reduces, the convergence becomes slower. This can be seen across models and datasets as shown in tables 3(a) and 4(a). As an example, table 4 shows that with $10000\times$ compression, we require 4.55 epochs as compared to the original model's one epoch. This might seem unfavorable at first. However, as seen in the previous section, there is a significant gain in training time per iteration. As seen in table 4, this gain largely compensates for the disadvantage in terms of convergence, making the overall training times similar. For example, while we need 4.55 epochs with $10000\times$ compressed PSS, the training time per iteration goes down by a factor of 4.37. Hence, the overall training time is only $1.04\times$ original time.

**Engineering challenges and costs vs. excess parameters:** The excess parameters in a recommendation model primarily appear due to the construction of embedding tables. The industrial-scale embedding tables can go as large as hundreds of terabytes. With the increase in the model's size, the model needs to be distributed across different nodes and GPUs. The complexities of efficiently running a distributed model training include many considerations, such as non-uniform memory allocation and communication costs. The increasing literature detailing the engineering solutions to training such models is evidence of the fact that training such models require engineering ingenuity to solve the challenges involved [3, 25, 26, 27, 28, 29]. For example, in [3], the authors detail their solution to train a model of size 50TB on a new distributed system. Similarly, in [25], authors talk about their optimizations of recommendation models on CPU clusters. In this paper, we argue that the large-scale nature of embedding-based recommendation models appears due to embedding tables where a complete $n \times d$ table with blown up $n$ does not add value to model capacity while adding significant engineering challenges. Distributed systems also imply significant energy costs. PSS can avoid these downsides by fitting entire learnable parameters on a single GPU/node.

**Discussion on the model-training speed with different methods.** Most of the other compression techniques, such as pruning, low-precision quantized embeddings, TT-REC, and MD embeddings, are limited in the amount of compression they offer. For instance, methods like pruning, TT-REC, and MD embeddings cannot offer a compression more than $d\times$ for an embedding table $E : n \times d$. Low-precision embeddings cannot offer compression of more than $32\times$. Thus, for industrial scale models where n is very large, it is unlikely that these compression techniques will be able to eliminate model-parallel training. Parameter sharing methods such as hashing trick and QR embeddings, while they can give arbitrary compression in theory, have failed to maintain quality beyond $4\times$ compression and thus cannot eliminate model-parallel training. In the case of criteo-tb embedding compression in literature, TT-REC gives the compression of $117\times$. Paraphrasing from TT-REC [16], they are able to

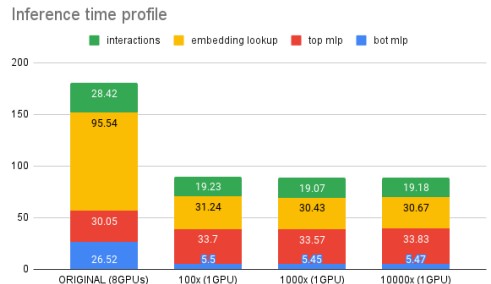

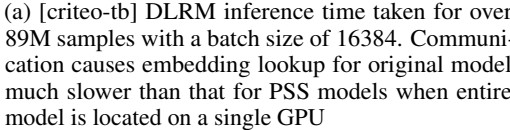

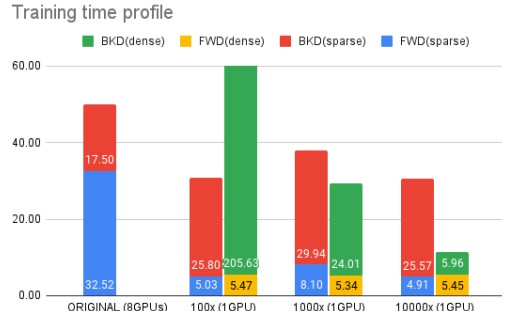

(a) [criteo-tb] DLRM inference time taken for over 89M samples with a batch size of 16384. Communication causes embedding lookup for original model much slower than that for PSS models when entire model is located on a single GPU

(b) DLRM training time for 1000 iterations of training with a mini-batch size of 2048 and SGD optimizer

train a $37.4\times$ compressed models in $1.11\times$ time. While, with our PSS using ROBE-Z embeddings, we are able to train a $10000\times$ compressed model with $1.04\times$ time.

## 5    Limitations and future work

This paper shows that we can compress the embedding tables for CTR prediction models as much as $10000\times$. However, CTR is a specific recommendation problem with the binary output. It would be interesting to see if these compression rates hold for other recommendation-related scenarios, such as ranking or retrieval. The theory presented in the paper is general, and hence, it is promising for extensions of this work to other recommendation problems or otherwise. However, we should note that the theory only evaluates the effect of embedding tables on the first layer (dot product with a vector $x \in R^d$) of the model and does not talk about its impact on the entire recommendation model. At this stage, it is difficult, if not impossible, to analyze the effect of embedding compression over the whole recommendation model. We propose to extend compressed embeddings to other recommendation problems and theories for compressed embeddings in the future.

## 6    Conclusion

Today's prevalent idea in deep learning(DL) is overparameterized models with stochastic iterative learning. While this paradigm has led to success in DL, the sustainability of this route to success is a grim question [19]. We evaluate the value of embedding-based excess parameters in DLRM models. We conclude that compressed DLRM models are better for fast inference and are easy to train and deploy as expected. More importantly, they can also be trained in the same overall time despite having a slower convergence rate. The critical observation is that compressed models enjoy a system advantage they can exploit, which reduces time per iteration. The paper also highlights that the correct way to compare the quality of models of highly different sizes is to run the models for equal time rather than equal iterations.

## 7    Acknowledgements

This work was supported by National Science Foundation SHF-2211815, BIGDATA-1838177, ONR DURIP Grant, and grants from Adobe, Intel, Total, and VMware. We also thank Ben Coleman for giving helpful writing comments on the paper.

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
