# A   Code Links

- ROBE-Z patches : https://github.com/apd10/universal_memory_allocation/tree/paddingidx_sparse
- DLRM criteo-tb code (patch over original facebook code) : https://github.com/apd10/dlrm
- criteo-kaggle code link https://github.com/apd10/criteo_deepctr

# B   Theorem 6

*Theorem:* Let the embedding table be $E \in R^{n \times d}$. Consider a matrix $S$ which is $JLT(\epsilon, \delta, 9^d)$). Then $M$ and $\mathcal{M}$ defined by

$$M = SE \quad \mathcal{M}(M, i) = (Se_i)^\top (M)$$

is a Macro$(\epsilon, \delta)$-PSS.

*Proof.* In this section we want to analyze the embedding table as a whole under compression. Consider the embedding table $E$. Generally, with deep learning, the embeddings taken from the embedding table undergo a functional transformation. If we only focus on only the first layer, we would usually see an operation like the following for some i, x

$$e_i^T E x \tag{2}$$

Let us restrict ourselves to this computation. Also, let us look at all the embeddings at once. So for some $x \in R^d$ let us consider the following vector

$$E x \tag{3}$$

as a vector that encapsulates the effect of $E$ on this $x$ (which is the first layer of the model. Let $S : k \times n$ be the l2-subspace embedding matrix [20]. Then, leveraging the known result in randomized linear algebra, we know that ,

$$\|(SE)x\|_2 = (1 \pm \epsilon)\|Ex\|_2 \; \forall x \in R^d \tag{4}$$

This translates to

$$\langle (SE)x, (SE)y \rangle = \langle Ex, Ey \rangle \pm \mathcal{O}(\epsilon) \forall x, y \in R^d \; \|x\| = \|y\| = 1 \tag{5}$$

Note that (SE) is the proposed compressed representation of the embedding table and is of size $k \times d, k \ll n$. There are various types of l2-subspace embedding matrices. Specifically, the class of $JLT(\epsilon, \delta, 9^d)$ matrices are l2-subspace embedding matrices. We know that we can build JLT matrix of size $k \times n$ where $k = \Theta((d + log(1/\delta))\epsilon^{-2})$. Thus, in terms of obtaining a $\epsilon, \delta$ approximation for embedding table, we only need $\mathcal{O}(d(d + log(1/\delta)/\epsilon^2))$ parameters to approximate the column subspace of E , $Ex \forall x \in R^d$. $\qquad \square$

# C   Theorem 7

*Theorem:* Let the embedding table be $E \in R^{n \times d}$. Consider a matrix $S$ which is $JLT(\rho\epsilon, \delta, 9^d + n)$) Then $M$ and $\mathcal{M}$ defined by

$$M = SE \quad \mathcal{M}(M, i) = (Se_i)^\top (M)$$

is a Micro $(\epsilon, \delta, \rho)PSS$ where $e_i \in R^n$ is a one-hot encoding of integer $i$ (i.e $e_i[i] = 1$ and rest all elements of $e_i$ are 0).

*Proof.* Johnson Lindenstrauss transforms are a class of transformations which preserve pair wise inner products ( equivalently norms ) of a set of vectors in a space.

Definition of JLT transform from [20] is as follows: random matrix $S \in R^{k \times n}$ is a $JLT(\epsilon, \delta, f)$ if with probability $(1 - \delta)$, for any f-element subset $V \subset R^n$, for all $v_1, v_2 \in V$, it holds that

$$|\langle Sv_1, Sv_2 \rangle - \langle v_1, v_2 \rangle| \leq \epsilon \|v_1\|_2 \|v_2\|_2 \tag{6}$$

Consider the following sets of points.

1. $\{e_i\}_{i=0}^{n-1}$

2. $1/2 - net$ over the set $\{u | u \in \text{Column-space}(E) \; \|u\|\|_2 = 1\}$

We use the lemma 5 from [20] to have the number of points in set 2 bounded by $9^d$. Let S be a matrix that is JLT$(\delta, \epsilon, (n + 9^d))$. For these $(n + 9^d)$ points with probability $(1 - \delta)$, the matrix S preserves inner products as per definition of JLT given above.

**part A:** We will show that using the matrix S, for all $v_1, v_2 \in \{e_i\} \cup \text{column-space}(E)$ it holds that ,

$$|\langle Sv_1, Sv_2 \rangle - \langle v_1, v_2 \rangle| \le \epsilon \|v_1\|_2 \|v_2\|_2 \tag{7}$$

Let us consider 3 cases.

1. $v_1, v_2 \in \text{Column-space}(E)$. We use the same argument as given on page 12 [20] and conclude that condition holds for these $v_1, v_2$

2. $v_1, v_2 \in \{e_i\}_{i=0}^{n-1} \cup 1/2 - net$. condition holds due to JLT

3. $v_1 \in \{e_i\}_{i=0}^{n-1}, v_2 \in \text{column-space}(E)$. Let $\|v_2\| = 1$. We will prove the condition for this case and for all other non-unit-norm cases will happen due to scaling. Using argument from page 12 [20], we can represent $v_2$ as a sum of vectors in $1/2 - net$.

$$v_2 = v^0 + v^1 + v^2 + .... \quad \text{where } v^i \in 1/2\text{-net} \tag{8}$$

such that $\|v^i\| \le \frac{1}{2^i}$

$$|\langle Sv_1, Sv_2 \rangle - \langle v_1, v_2 \rangle|$$
$$= |\sum_i (\langle Sv_1, Sv^i \rangle) - \sum_i (\langle v_1, v^i \rangle)|$$
$$= |\sum_i (\langle Sv_1, Sv^i \rangle) - (\langle v_1, v^i \rangle)|$$
$$\le \sum_i (\langle Sv_1, Sv^i \rangle) - (\langle v_1, v^i \rangle)|$$
$$\le (\epsilon) \sum_i \|v_1\|_2 \|v^i\|_2$$
$$= \epsilon \sum_i \|v^i\|_2$$
$$\le \epsilon \sum_i \frac{1}{2^i}$$
$$= 2\epsilon$$

Thus, using the matrix S, for all $v1, v2 \in \{e_i\} \cup \text{column-space}(E)$ it holds that

$$|\langle Sv_1, Sv_2 \rangle - \langle v_1, v_2 \rangle| \le \epsilon \|v_1\|_2 \|v_2\|_2 \tag{9}$$

**Part B:** Proving the theorem for using S to create $(\epsilon, \delta, \rho)$ PSS

Let $e_i$s be one hot encoded vectors such that we have that

$$E[i] = e_i^\top E \tag{10}$$

Let $M = SE$ and $\mathcal{M}(M, i) = (SE)^\top Se_i$

$$\langle \mathcal{M}(M, i), x \rangle - \langle E[i]^\top, x \rangle \tag{11}$$
$$= \langle (SE)^\top Se_i, x \rangle - \langle E[i]^\top, x \rangle \tag{12}$$
$$= \langle Se_i, SEx \rangle - \langle (e_i^\top E)^\top, x \rangle \qquad = \langle Se_i, SEx \rangle - \langle e_i, Ex \rangle \tag{13}$$

Note that both $e_i$ and $Ex$ are vectors that belong to our f point set V. Thus

$$\langle \mathcal{M}(M,i), x \rangle - \langle E[i]^\top, x \rangle \tag{14}$$

$$= \langle Se_i, SEx \rangle - \langle e_i, Ex \rangle \tag{15}$$

$$\leq \epsilon \|e_i\|_2 \|Ex\|_2 \tag{16}$$

$$\leq \epsilon \|Ex\|_2 \tag{17}$$

$$\leq \epsilon \sigma(E) \|x\|_2 \tag{18}$$

$$\leq \epsilon \frac{\sigma(E)}{\|E[i]\|_2} \|E[i]\|_2 \|x\|_2 \tag{19}$$

$$\leq \epsilon \frac{1}{\rho(i)} \|E[i]\|_2 \, \|x\|_2 \tag{20}$$

Note that $\|e_i\| = 1$ Let $\sigma$ be the max singular value of E. If we use $JLT(\rho\epsilon, \delta, 9^d + n)$, then we will have,

$$\langle \mathcal{M}(M,i), x \rangle - \langle E[i]^\top, x \rangle \tag{21}$$

$$\leq \epsilon \frac{\rho}{\rho(i)} \|E[i]\|_2 \, \|x\|_2 \tag{22}$$

Thus if $\rho(i) > \rho$ then

$$\langle \mathcal{M}(M,i), x \rangle - \langle E[i]^\top, x \rangle \tag{23}$$

$$\leq \epsilon \|E[i]\|_2 \, \|x\|_2 \tag{24}$$

Hence, proved.

$\square$

# D   Proof of theorem 4

Let $(M, \mathcal{M})$ be a Micro $(\epsilon, \delta, \rho)$-PSS for embedding table E, we have ,

$$\forall i, j \in \{0, ..., n{-}1\}, \text{s.t } \rho(i), \rho(j) > \rho$$
$$|\,|\langle \mathcal{M}(M,i), \mathcal{M}(M,j) \rangle - \langle E[i], E[j] \rangle| \leq (\epsilon^2 + 2\epsilon) \|E[i]\|_2 \|E[j]\|_2$$

*Proof.* **part A:** we will first bound the $\|\mathcal{M}\|_2$ in terms of $\|E[i]\|_2$

Using definition of PSS

$$|\langle \mathcal{M}(M,i), \mathcal{M}(M,i) \rangle - \langle \mathcal{M}(M,i), E[i] \rangle| \leq \epsilon \|\mathcal{M}(M,i)\|_2 \|E[i]\|_2$$

$$\langle \mathcal{M}(M,i), \mathcal{M}(M,i) \rangle \leq \langle \mathcal{M}(M,i), E[i] \rangle + \epsilon \|\mathcal{M}(M,i)\|_2 \|E[i]\|_2$$
$$\text{and } \langle \mathcal{M}(M,i), \mathcal{M}(M,i) \rangle \geq \langle \mathcal{M}(M,i), E[i] \rangle - \epsilon \|\mathcal{M}(M,i)\|_2 \|E[i]\|_2$$

$$\langle \mathcal{M}(M,i), \mathcal{M}(M,i) \rangle \leq \langle E[i], E[i] \rangle + \epsilon (\|E[i]\|_2^2 + \|\mathcal{M}(M,i)\|_2 \|E[i]\|_2)$$
$$\text{and } \langle \mathcal{M}(M,i), \mathcal{M}(M,i) \rangle \geq \langle E[i], E[i] \rangle - \epsilon (\|E[i]\|_2^2 + \|\mathcal{M}(M,i)\|_2 \|E[i]\|_2)$$

Let $\|\mathcal{M}(M,i)\| = m$ and $\|E[i]\| = e$

$$m^2 - \epsilon em \leq e^2 + \epsilon e^2 \text{ and } m^2 + \epsilon em \geq e^2 - \epsilon e^2$$

Adding $(1/4)\epsilon^2 e^2$ on both sides

$$m^2 - \epsilon em + \frac{1}{4}\epsilon^2 e^2 \leq e^2 + \epsilon e^2 + \frac{1}{4}\epsilon^2 e^2 \text{ and } m^2 + \epsilon em + \frac{1}{4}\epsilon^2 e^2 \geq e^2 - \epsilon e^2 + \frac{1}{4}\epsilon^2 e^2$$

$$(m - 1/2\epsilon e)^2 \leq (e + 1/2\epsilon e)^2 \text{ and } (m + 1/2\epsilon e)^2 \geq (e + 1/2\epsilon e)^2$$

As both m and e are positive.

$$m \leq (e + \epsilon e) \text{ and } m \geq e - \epsilon e$$

Thus,

$$|m - e| \leq \epsilon e$$

Thus,

$$\|\mathcal{M}(M, i)\|_2 = (1 \pm \epsilon)\|E[i]\|_2 \tag{25}$$

**Part B:** Now we can look at the pairwise inner products.

$$|\langle \mathcal{M}(M, i), \mathcal{M}(M, j)\rangle - \langle E[i], E[j]\rangle|$$
$$\leq |\langle \mathcal{M}(M, i), \mathcal{M}(M, j)\rangle - \langle E[i], \mathcal{M}(M, j)\rangle| + |\langle \mathcal{M}(M, j), E[i]\rangle - \langle E[j], E[i]\rangle|$$
$$\leq \epsilon(\|E[i]\|_2\|\mathcal{M}(M, j)\|_2) + \epsilon\|E[i]\|_2\|E[j]\|_2$$
$$\leq \epsilon(1 + \epsilon)(\|E[i]\|_2\|E[j]\|_2) + \epsilon\|E[i]\|_2\|E[j]\|_2$$
$$= \epsilon(2 + \epsilon)(\|E[i]\|_2\|E[j]\|_2)$$

Hence, proved $\qquad\qquad\square$

# E Data

## E.1 PSS rigorous evaluation

- Page 1 : memory vs compression.
- page 2: memory vs chunk size
- page 3: compression vs chunk size

Some obervations

- (sparse/dense) original embeddings should be run with sparse gradients. The dense embeddings are too time consuming as they require updating large amounts of memory (vacuously) in each iteration. Note that a lot of optimizers in deep learning libraries like pytorch do not yet have support for sparse gradient updates.
- When the entire embedding table can fit on the GPU, the forward and backward times do not change much.
- While we can go upto 64M tokens ( for m = 128 ) on single gpu using original embeddings and sparse gradient propogation, the simulated embedding tables can be much larger with PSS and compression.
- Best forward times we can achieve with PSS ( around 0.27ms/iteration ) is almost 2× the forward times for original embedding lookup. (0.12 ms / iteration). This is expected since, we have to perform additional hash computations in PSS. Hence, if both original embeddings and compressed embeddings are at the same distance from the computational resource, then PSS will have a disadvantage.
- (compression, PSS) Higher the compression, better are timings for both backward pass (using dense propagation) and forward pass.
- A nice trade-off can be seen between using sparse or dense gradients with PSS. If we use dense gradients, the time in backward propogation is affected by the overall size of embedding parameters. Thus, for higher compression and smaller $n$, we have smaller times where as if n becomes larger and compression is smaller, the time taken increases. The cost of dense gradient propogation becomes quite high at sufficiently large $n$ and sufficiently small compression. On the other hand, the cost of sparse gradient propogation is uniform across different values of $n$ and compression. This is because the algorithmic complexity and memory accessed is similar. This gives us a guideline as to when to use sparse / dense gradients with PSS. Note that in our final results, we see a good overall improvement by leveraging the dense gradient propogation at high compression.
- Higher the chunk size better is the time in forward pass. backward pass is largely unaffected by chunk size largely due to the way it is implemented.

| Original time(ms) | | | | | | | |
|---|---|---|---|---|---|---|---|
| | | | | n | | | |
| **call** | **sparse** | 4M | 8M | 16M | 32M | 64M | 128M |
| backward | FALSE | 25.68 | 50.72 | 100.72 | 200.67 | | |
| | TRUE | 0.35 | 0.35 | 0.38 | 0.38 | 0.35 | |
| forward | FALSE | 0.21 | 0.22 | 0.22 | 0.22 | | |
| | TRUE | 0.12 | 0.12 | 0.12 | 0.12 | 0.12 | |

Table 5: Time taken for original embeddingn table

| Forward pass (ms) [sparse=false] | | | | |
|---|---|---|---|---|
| | compression | | | |
| n | 10x | 100x | 1000x | 10000x |
| 4M | 0.40 | 0.31 | 0.27 | 0.27 |
| 8M | 0.36 | 0.32 | 0.29 | 0.27 |
| 16M | 0.37 | 0.31 | 0.30 | 0.27 |
| 32M | 0.37 | 0.25 | 0.31 | 0.26 |
| 64M | 0.37 | 0.37 | 0.30 | 0.28 |
| 128M | 0.37 | 0.37 | 0.32 | 0.29 |
| 256M | | 0.36 | 0.31 | 0.30 |

Table 6: batch size = 10240 and dimension = 128

| Forward pass (ms) [sparse=true] | | | | |
|---|---|---|---|---|
| | compression | | | |
| n | 10x | 100x | 1000x | 10000x |
| 4M | 0.57 | 0.54 | 0.34 | 0.33 |
| 8M | 0.51 | 0.56 | 0.34 | 0.32 |
| 16M | 0.56 | 0.56 | 0.36 | 0.34 |
| 32M | 0.55 | 0.57 | 0.36 | 0.33 |
| 64M | 0.56 | 0.57 | 0.52 | 0.35 |
| 128M | 0.52 | 0.57 | 0.57 | 0.35 |
| 256M | 0.55 | 0.53 | 0.51 | 0.36 |

Table 7: batch size = 10240 and dimension = 128

| Backward pass (ms) [sparse=false] | | | | |
|---|---|---|---|---|
| | compression | | | |
| n | 10x | 100x | 1000x | 10000x |
| 4M | 2.55 | 0.59 | 0.35 | 0.35 |
| 8M | 4.81 | 0.83 | 0.37 | 0.35 |
| 16M | 9.15 | 1.27 | 0.41 | 0.34 |
| 32M | 17.81 | 2.13 | 0.54 | 0.35 |
| 64M | 35.10 | 3.85 | 0.73 | 0.36 |
| 128M | 69.73 | 7.42 | 1.08 | 0.40 |
| 256M | | 14.34 | 1.78 | 0.47 |

Table 8: batch size = 10240 and dimension = 128

| Backward pass (ms) [sparse=true] | | | | |
|---|---|---|---|---|
| | compression | | | |
| n | 10x | 100x | 1000x | 10000x |
| 4M | 2.67 | 2.48 | 2.56 | 2.52 |
| 8M | 2.53 | 2.58 | 2.48 | 2.53 |
| 16M | 2.52 | 2.68 | 2.58 | 2.45 |
| 32M | 2.51 | 2.64 | 2.56 | 2.55 |
| 64M | 2.51 | 2.66 | 2.63 | 2.60 |
| 128M | 2.52 | 2.65 | 2.65 | 2.54 |
| 256M | 2.54 | 2.52 | 2.53 | 2.62 |

Table 9: batch size = 10240 and dimension = 128

| Forward pass (ms) [sparse=false] | | | | | | |
|---|---|---|---|---|---|---|
| | PSS chunk size | | | | | |
| n | 1 | 2 | 4 | 8 | 16 | 32 |
| 4M | 0.68 | 0.58 | 0.55 | 0.32 | 0.31 | 0.31 |
| 8M | 0.75 | 0.62 | 0.54 | 0.33 | 0.31 | 0.31 |
| 16M | 0.77 | 0.65 | 0.54 | 0.54 | 0.31 | 0.31 |
| 32M | 0.80 | 0.67 | 0.54 | 0.26 | 0.24 | 0.23 |
| 64M | 0.92 | 0.79 | 0.65 | 0.39 | 0.37 | 0.37 |
| 128M | 0.88 | 0.75 | 0.65 | 0.39 | 0.37 | 0.37 |
| 256M | 0.88 | 0.75 | 0.65 | 0.40 | 0.36 | 0.36 |

Table 10: batch size = 10240 and dimension = 128

| Forward pass (ms) [sparse=true] | | | | | | |
|---|---|---|---|---|---|---|
| | PSS chunk size | | | | | |
| $n$ | 1 | 2 | 4 | 8 | 16 | 32 |
| 4M | 0.87 | 0.62 | 0.57 | 0.57 | 0.52 | 0.37 |
| 8M | 0.91 | 0.76 | 0.70 | 0.71 | 0.57 | 0.57 |
| 16M | 0.94 | 0.80 | 0.68 | 0.70 | 0.56 | 0.56 |
| 32M | 0.92 | 0.82 | 0.68 | 0.69 | 0.57 | 0.54 |
| 64M | 0.93 | 0.80 | 0.68 | 0.67 | 0.56 | 0.57 |
| 128M | 0.93 | 0.80 | 0.68 | 0.67 | 0.56 | 0.56 |
| 256M | 0.93 | 0.80 | 0.68 | 0.67 | 0.55 | 0.37 |

Table 11: batch size = 10240 and dimension = 128, fixed compression = 100x

| Backward pass (ms) [sparse=false] | | | | | |
|---|---|---|---|---|---|
| | **PSS chunk size** | | | | |
| *n* | 1 | 2 | 4 | 8 | 16 | 32 |
| 4M | 1.29 | 1.10 | 0.80 | 0.64 | 0.58 | 0.57 |
| 8M | 1.56 | 1.38 | 1.07 | 0.88 | 0.82 | 0.80 |
| 16M | 2.02 | 1.87 | 1.54 | 1.33 | 1.27 | 1.27 |
| 32M | 2.96 | 2.79 | 2.40 | 2.20 | 2.12 | 2.12 |
| 64M | 4.69 | 4.54 | 4.13 | 3.98 | 3.85 | 3.85 |
| 128M | 8.15 | 8.00 | 7.70 | 7.48 | 7.43 | 7.42 |
| 256M | 15.17 | 15.01 | 14.61 | 14.40 | 14.34 | 14.34 |

Table 12: batch size = 10240 and dimension = 128, fixed compression = 100x

| Backward pass (ms) [sparse=true] | | | | | |
|---|---|---|---|---|---|
| | **PSS chunk size** | | | | |
| *n* | 1 | 2 | 4 | 8 | 16 | 32 |
| 4M | 3.17 | 3.04 | 2.61 | 2.54 | 2.50 | 2.48 |
| 8M | 3.41 | 2.97 | 2.64 | 2.64 | 2.66 | 2.61 |
| 16M | 3.56 | 3.07 | 2.74 | 2.58 | 2.54 | 2.62 |
| 32M | 3.68 | 3.13 | 2.73 | 2.63 | 2.53 | 2.50 |
| 64M | 3.94 | 3.36 | 2.91 | 2.64 | 2.52 | 2.64 |
| 128M | 3.98 | 3.22 | 2.77 | 2.60 | 2.51 | 2.60 |
| 256M | 4.00 | 3.27 | 2.74 | 2.61 | 2.53 | 2.50 |

Table 13: batch size = 10240 and dimension = 128, fixed compression = 100x

| Forward pass (ms) [sparse=false] | | | | |
|---|---|---|---|---|
| | **compression** | | | |
| **chunk** | 10x | 100x | 1000x | 10000x |
| 1 | 0.89 | 0.90 | 0.73 | 0.28 |
| 2 | 0.77 | 0.80 | 0.61 | 0.27 |
| 4 | 0.65 | 0.65 | 0.54 | 0.27 |
| 8 | 0.40 | 0.39 | 0.32 | 0.28 |
| 16 | 0.37 | 0.37 | 0.30 | 0.29 |
| 32 | 0.37 | 0.37 | 0.30 | 0.30 |

Table 14: batch size = 10240 and dimension = 128

| Forward pass (ms) [sparse=true] | | | | |
|---|---|---|---|---|
| | **compression** | | | |
| **chunk** | 10x | 100x | 1000x | 10000x |
| 1 | 0.93 | 0.93 | 0.87 | 0.34 |
| 2 | 0.80 | 0.80 | 0.77 | 0.32 |
| 4 | 0.69 | 0.68 | 0.71 | 0.33 |
| 8 | 0.69 | 0.67 | 0.72 | 0.34 |
| 16 | 0.56 | 0.57 | 0.49 | 0.34 |
| 32 | 0.56 | 0.55 | 0.54 | 0.36 |

Table 15: batch size = 10240 and dimension = 128

| Backward pass (ms) [sparse=false] | | | | |
|---|---|---|---|---|
| | compression | | | |
| chunk | 10x | 100x | 1000x | 10000x |
| 1 | 35.94 | 4.69 | 1.44 | 0.37 |
| 2 | 35.78 | 4.53 | 1.26 | 0.37 |
| 4 | 35.38 | 4.13 | 0.96 | 0.36 |
| 8 | 35.15 | 3.91 | 0.83 | 0.36 |
| 16 | 35.10 | 3.85 | 0.72 | 0.36 |
| 32 | 35.11 | 3.86 | 0.70 | 0.37 |

Table 16: batch size = 10240 and dimension = 128

| Backward pass (ms) [sparse=true] | | | | |
|---|---|---|---|---|
| | compression | | | |
| chunk | 10x | 100x | 1000x | 10000x |
| 1 | 4.08 | 3.89 | 2.77 | 2.87 |
| 2 | 3.32 | 3.20 | 2.76 | 2.88 |
| 4 | 2.78 | 2.78 | 2.61 | 2.63 |
| 8 | 2.62 | 2.60 | 2.58 | 2.58 |
| 16 | 2.51 | 2.68 | 2.63 | 2.55 |
| 32 | 2.49 | 2.50 | 2.59 | 2.57 |

Table 17: batch size = 10240 and dimension = 128