# OpenReview forum: "The trade-offs of model size in large recommendation models : 100GB to 10MB Criteo-tb DLRM model"
_NeurIPS.cc/2022/Conference — NeurIPS 2022 Accept_

### Official Review · Reviewer_HDZP · 2022-07-11

**Rating:** 6
**Confidence:** 3
**Soundness:** 4 excellent
**Presentation:** 3 good
**Contribution:** 2 fair

**Summary:**

This paper focuses on the problem of compressing large embedding tables in recommender systems. The authors provide a theoretical analysis of the memory requirements and approximation error of compressing embedding tables under the defined parameter sharing setup. They also show the increase of compression rate accelerates the training of each iteration while leading to more iterations until model convergence. The conclusion is that compressed models can be trained in the same overall time and are faster for inference compared with uncompressed ones.

**Questions:**

Q1. Please see W1, what do you think of it?

Q2. Please see W2, what are the existing implementation choices for PSS methods?

Q3. Please see W3, can you summarize the main contribution of the paper?

**Limitations:**

Yes, the authors have adequately addressed the limitations and potential negative societal impact of their work.

**Strengths And Weaknesses:**

S1. This paper is mostly well-organized and clearly presented. Figures are helpful, e.g. Figure 1 does help to clarify the tradeoffs of training larger embedding tables.

S2. The authors conduct experiments on real-world datasets. The results are sufficient to show the compensation of faster training speed for slower convergence of compressed models.

S3. The conclusions about compressed models are intuitive and are beneficial to the application of compressed models in real systems.

W1. The theoretical analysis of the memory requirements and approximation error is based on the defined parameter sharing setup. It is not clear whether parameter-sharing methods have become state-of-the-art methods of embedding compression. Thus the impact of the conclusions may be limited.

W2. Section 3 introduces several implementation choices for PSS, but the originality is not clear since the choices used by existing PSS methods are not mentioned.

W3. Due to the above two points, the contribution of this paper is not quite substantial and should be better summarized.

---

> ### Author Response · Authors · 2022-08-02
> **reply to the comments**
>
> 1. “W1. The theoretical analysis of the memory requirements and approximation error is based on the defined parameter sharing setup. It is not clear whether parameter-sharing methods have become state-of-the-art methods of embedding compression. Thus the impact of the conclusions may be limited “
>
> There is a line of research on compressing embedding tables in literature since DLRM paper which showcased the problem of large embedding tables. Criteo datasets have been the gold standard for measuring these approaches. To the best of our knowledge parameter sharing approaches have shown the state-of-the art results with ROBE. Also, 3 out of 5 methods explored for embedding compression are parameter sharing based. ( hashing trick, QR embeddings, HashedNet / ROBE). Thus, we believe that, if not common-place, they are certainly a major direction of research in embedding compression.
>
> 2. “W2. Section 3 introduces several implementation choices for PSS, but the originality is not clear since the choices used by existing PSS methods are not mentioned.”
>
> Section 3 shows what implementation choices are made to make PSS achieve higher compression, faster training time per iteration to obtain a win-win situation : similar quality, faster inference and overall similar training. These implementation choices are relevant for high compression setups. We take existing ROBE implementation and apply these choices to improve its performance.
>
> Existing PSS:
>
> Hashing Trick and QR Embeddings : This uses native embedding tables which will best run with sparse gradients due to larger overall memory usage.
>
> HashedNet and ROBE-Z : They can achieve high compression. HashedNet does not use chunks, whereas ROBE-Z uses  chunks for  hashing. We use chunks and further improve implementation by analyzing the effect of sparse and dense gradients and tuning kernels.
>
> 3. “W3. Due to the above two points, the contribution of this paper is not quite substantial and should be better summarized.”
>
> Contributions:
>
> A. We provide a theoretical insight into why parameter sharing approaches perform well for embedding compression.
>
> a. We discover that we can expect  very high compression for embedding tables (O(log(n))
>
> b. We also learn that larger embedding tables can be compressed more.
>
> c. Also, the theory shows promise to generalize these results obtained on CTR models to other domains and problems such as ranking/retrieval.
>
> To the best of our knowledge, this is the first analysis which analyzes a compression of an entire embedding table as opposed to previous methods which either do not provide theoretical analysis or restrict the analysis to inner product preservation.
>
> B. We provide interesting trade-off analysis for state of the art DLRM model on industry standard criteo dataset and show that compression embedding tables is a win-win situation. While faster inference is free with compression, training convergence is previously shown to be slower. However, with good choices of implementation, we can show that the system benefit of smaller models can compensate for slower convergence and we can essentially have similar overall training times.

---

### Official Review · Reviewer_XHj9 · 2022-07-11

**Rating:** 5
**Confidence:** 3
**Soundness:** 3 good
**Presentation:** 2 fair
**Contribution:** 3 good

**Summary:**

The paper studies how random projections on embedding matrices could lead to heavy compressions of practical deep models. They picked a well-associable Criteo-TB dataset with binary loss functions and discovered that most SOTA baselines such as DLRM could be compressed by 10k times without observable performance degradation. The paper also argue that previous failures on random projection could be due to a lack of sufficient training epochs. Some theoretical background was introduced as well as a few other papers on related implementation details.

**Questions:**

Sections 2.1-2.3 on theoretical analysis exist without a purpose. Many things should be discussed. For example:
* Line 107. Norm of recovered embedding from PSS. Why does it matter to have l2-loss in addition to inner-product loss? Do you expect the embedding vectors to be unit-normed?
* Line 123. The appearance of singular value is not motivated. Would scale of the embedding vectors matter? What actual values do you observe in trained models?
* In general, what actual knowledge could a reader gain besides saying that "random projection in general is a good idea"?

I did not go into the details of Section 2.5. According to the authors, the most successful model runs without sparse hashes, so I don't know how I should appreciate Sparse-JL. Also, what does QR decomposition do here?

**Limitations:**

The paper did not discuss limitations beyond the presented variants of a single dataset. I would suspect that random projection may not generally work well in retrieval-type of problems, such as search and masked language modeling.

**Strengths And Weaknesses:**

Strengths:
* Originality: While Johnson Lindenstrauss Transform tells us that random projections from sparse vectors could just work, showing that the simple idea actually works is a good piece of knowledge to have.
* Originality: The paper is based on well-known datasets and SOTA methods. The results appear associable.

Weaknesses:
* Significance: In my experience, random projections do not tend to work well in retrieval problems with cross-entropy loss. This paper is only limited to binary prediction losses in some variants of the same dataset. The authors may want to discuss limitations of their work, esp. with respect to retrieval problems such as BPR loss in recommender systems or XE loss in masked-language modeling.

---

> ### Author Response · Authors · 2022-08-02
> **reply to the comments**
>
> 1. “Significance: In my experience, random projections do not tend to work well in retrieval problems with cross-entropy loss. This paper is only limited to binary prediction losses in some variants of the same dataset. The authors may want to discuss limitations of their work, esp. with respect to retrieval problems such as BPR loss in recommender systems or XE loss in masked-language modeling.”
>
> - We added a pace holder limitations section in the paper which will be elaborate upon in the final version ( see blue section at the end ) . In this paper, we show that we can compress the embedding tables for CTR prediction models as much as 10000$\times$. The theory presented in the paper shows promise of generalizing these results to other models such as retrieval or ranking problems. However, more experiments are needed to evaluate PSS approaches in these problems. We evaluate CTR models in this paper and extensions of this work to other models is left for future work.
> - We believe that learning compressed representations motivated by random projections might perform better than applying random projections as a compression technique on learned embedding tables. So we are optimistic about applying learned PSS to retrieval and ranking problems.
>
> 2. “Sections 2.1-2.3 on theoretical analysis exist without a purpose”.
>
> While we agree that generic theoretical analysis of dimensionality reduction provides limited utility in practice. This is exactly the kind of gap we tried to fill in. Our theoretical analysis is not merely a dimensionality reduction but catered to the compression and reconstruction, instead of just inner product (or norm preservation) which is there in the standard literature. We believe our analysis provided the following.
> - It provides explanations as to why methods like ROBE have shown impressive compression results.
> - It gives us an idea of how much compression we can expect. Answers questions like can we expect higher compression for larger embedding tables.
> - It also evaluates if the results are generalizable to other settings and models.
>
> 3. "Line 107. Norm of recovered embedding from PSS. Why does it matter to have l2-loss in addition to inner-product loss? Do you expect the embedding vectors to be unit-normed?"
>
> It is standard in theory associated with random projections to analyze the norms under projection operation , the result of which can be extended to inner products seamlessly. The analysis does not depend on the norms of embedding vectors.
>
> 4. "Line 123. The appearance of singular value is not motivated. Would the scale of the embedding vectors matter? What actual values do you observe in trained models?"
>
> The analysis does not make any assumptions on the scale of the embedding vectors and hence the appearance of singular value in the expression. Trained models directly learn the compressed embedding representations and hence there is no original embedding table to compare to.
>
> 5. "In general, what actual knowledge could a reader gain besides saying that "random projection in general is a good idea"?"
>
> We believe that reader should note the following points,
> - Large compression of embedding tables using parameter shared setups is well motivated by theory and thus can be expected to generalize to other problems. Thus, it deserves exploration in other domains and problems  ( along the lines which you point out like - retrieval problems etc)
> - While large compression would work, it comes with its own set of problems such as slower convergence. We make a point that at a scale as large as that of industrial embedding tables, using compression leads to a win-win situation : it gives faster inference, simpler training setups, training time per iteration reduction compensates for the slow convergence leading to similar overall training times.
>
> 6. “I did not go into the details of Section 2.5. According to the authors, the most successful model runs without sparse hashes, so I don't know how I should appreciate Sparse-JL. "
>
> To the contrary, sparse hashes work well and we essentially say in section 2.5 that sparse hashes work well even after relaxing the independence assumption. This leads to very simple hashing schemes for PSS.
>
> 7. "Also, what does QR decomposition do here?”
>
> Section 2.5 looks back at different parameter sharing approaches and shows that they are instantiations of  general PSS setup.

---

### Official Review · Reviewer_foGx · 2022-07-12

**Rating:** 6
**Confidence:** 3
**Soundness:** 3 good
**Presentation:** 3 good
**Contribution:** 3 good

**Summary:**

This paper proposed a new compression method, Parameter Shared Setup (PSS), and its training method to compress the large-scale recommendation embedding table. The proposed new compression can achieve higher compression rate on larger dataset with almost no decrease in quality and training speed. It can achieve even 10000 × compression on criteo-tb DLRM model, which is much better than previous method. It provides both theoretical and experimental proof.



**Questions:**

As mentioned above, more speed experiment comparing with other baselines should be included.

**Limitations:**

The compression should consider how to increase the number of data point without retraining. That is also valuable especially for recommendation

**Strengths And Weaknesses:**

Strength.
1. The proposed method can compress the embedding table in memory of the order of O(logn). Larger dataset will benifit more from logn. Criteo-tb DLRM model can be compressed 10000 times.
2. Previous methods show that higher compression rate will suffer from logner training time. But in this paper, the larger number of epochs needed for convergence can be compensated by the faster iteration, which keeps the total training time comparable with others.
3. This paper provide both theoretical and experimental explanation for the performance.

Weakness.
The speed experiment needs more comparison  between different compression methods. Now the author only consider the comparison among different compression rate by their own method. More comparison is required to show the speed performance. The dense gradients should be effective on ROBE, if ROBE also requires training and can compress it 1000 times. Now the speed experiment only supports the longer epochs and faster single integration but can not show the advantage of the new methods.

---

> ### Author Response · Authors · 2022-08-02
> **Reply to the comments**
>
> 1. “The speed experiment needs more comparison between different compression methods.”
>
> The main point we want to make in the paper is that of trade-offs between model size vs training/inference times vs model quality. We use ROBE style hashing as our choice of PSS and show these trade-offs. We show that compressing embedding tables is a win-win situation especially in industrial scale embedding tables where system benefits of training on single GPU compensates for the convergence behavior of compressed embeddings. We believe that this is an important point to make for wider adoption of compressed embedding tables.
>
> Other compression methods and trade-offs.
>
> Pruning does not have a clear application to embeddings tables at-least at the compression we are interested in. For example, there is no way we can compress an embedding table of size NxD by dropping weights beyond “D”x without some embeddings having no weights.  It should be noted that our compression is even more than “D”x compression (less than one element per row)
>
> Quantization also cannot give a very large compression. At best, it can give a compression of 32x which implies 1 bit per value.
>
> Parameter sharing approaches such as QR-embeddings, hashing tricks are shown to only achieve lesser compression (order of 4x)  and hence have to be run using a model-parallel approach where embeddings are split across GPUs. Also, they perform additional computation when compared to original embeddings and hence would perform slower than original embedding
>
> Low-rank decomposition approaches such as MD embeddings again will have to be run across GPUs due to lower compression and hence would be slower than original. TT-REC, which achieves 100x compression for criteo-tb dataset, show in their paper that due to time taken by additional computations, TT-REC ( run on 1 GPU) has higher training time overheads with larger reduction in model sizes. Paraphrasing section 6.3 of the paper, going to 37.4x compression is associated with 11.8% increase in overall training time with optimal TT-REC configuration. In our experiments, we see that going to 10000x compression, we see only a 4% increase in overall training time. Also same as pruning, there is a limit of rank = 1 which sets an upper limit to compression of such methods. Essentially, we cannot go beyond one element per row (i.e. Dx compression for a NxD embedding table)
>
> (Proposed fix: Add a table to the paper measuring the effect of model size trade-offs for each of the above methods.)
>
> 2. “The compression should consider how to increase the number of data point without retraining. That is also valuable especially for recommendation”
>
> We are not sure what this means. Can you please elaborate on this point so that we can try to answer it.

---

### Meta-Review · Area_Chair_1TwE · 2022-08-27

**Recommendation:** Accept
**Confidence:** Certain

**Metareview:**

Overall, the paper is a good contribution to the model compression research, which achieved 10000x compression using parameter sharing, compared previous best 1000x. Compressing large models is an important direction and I believe this is a good progress. While there were some disagreements on the significance of the contribution, I feel the authors' response reasonably addressed them.

**Award:**

No

---

### Decision · Program_Chairs · 2022-09-14

Accept